# Investigation of the Effect of Exendin-4 on Oleic Acid-Induced Steatosis in HepG2 Cells Using Fourier Transform Infrared Spectroscopy

**DOI:** 10.3390/biomedicines10102652

**Published:** 2022-10-20

**Authors:** Olfa Khalifa, Kamal H. Mroue, Raghvendra Mall, Ehsan Ullah, Nayla S. Al-Akl, Abdelilah Arredouani

**Affiliations:** 1Diabetes Research Center, Qatar Biomedical Research Institute, Hamad Bin Khalifa University, Qatar Foundation, Doha 34110, Qatar; 2Qatar Environment and Energy Research Institute, Hamad Bin Khalifa University (HBKU), Qatar Foundation, Doha 34110, Qatar; 3Qatar Computing Research Institute, Hamad Bin Khalifa University, Qatar Foundation, Doha 34110, Qatar; 4Department of Immunology, St. Jude Children’s Research Hospital, 262 Danny Thomas Place, Memphis, TN 38105-3678, USA; 5College of Health and Life Sciences, Hamad Bin Khalifa University, Qatar Foundation, Doha 34110, Qatar

**Keywords:** non-alcoholic fatty liver disease, steatosis, Oil Red O staining, Fourier transform-infrared spectroscopy, principal component analysis

## Abstract

Non-alcoholic fatty liver disease (NAFLD) is a common liver lesion that is untreatable with medications. Glucagon-like peptide-1 receptor (GLP-1R) agonists have recently emerged as a potential NAFLD pharmacotherapy. However, the molecular mechanisms underlying these drugs’ beneficial effects are not fully understood. Using Fourier transform infrared (FTIR) spectroscopy, we sought to investigate the biochemical changes in a steatosis cell model treated or not with the GLP-1R agonist Exendin-4 (Ex-4). HepG2 cells were made steatotic with 400 µM of oleic acid and then treated with 200 nM Ex-4 in order to reduce lipid accumulation. We quantified steatosis using the Oil Red O staining method. We investigated the biochemical alterations induced by steatosis and Ex-4 treatment using Fourier transform infrared (FTIR) spectroscopy and chemometric analyses. Analysis of the Oil Red O staining showed that Ex-4 significantly reduces steatosis. This reduction was confirmed by FTIR analysis, as the phospholipid band (C=O) at 1740 cm^−1^ in Ex-4 treated cells is significantly decreased compared to steatotic cells. The principal component analysis score plots for both the lipid and protein regions showed that the untreated and Ex-4-treated samples, while still separated, are clustered close to each other, far from the steatotic cells. The biochemical and structural changes induced by OA-induced lipotoxicity are at least partially reversed upon Ex-4 treatment. FTIR and chemometric analyses revealed that Ex-4 significantly reduces OA-induced lipid accumulation, and Ex-4 also restored the lipid and protein biochemical alterations caused by lipotoxicity-induced oxidative stress. In combination with chemometric analyses, FTIR spectroscopy may offer new approaches for investigating the mechanisms underpinning NAFLD.

## 1. Introduction

Non-alcoholic fatty liver disease (NAFLD) refers to a spectrum of conditions, ranging from simple benign steatosis to more progressive non-alcoholic steatohepatitis (NASH), which can progress to fibrosis, then cirrhosis, and, in some cases, hepatocellular carcinoma [1]. NAFLD occurs when lipids, in the form of triglycerides (TGs), accumulate in more than 5% of hepatocytes, without excessive alcohol consumption, viral or autoimmune hepatitis, congenital hepatic disorders, or long-term use of steatosis-inducing medications [2]. In recent years, NAFLD has emerged as one of the world’s most common liver diseases [3] and is expected to become the primary cause of liver transplantation [4]. NAFLD affects 25% of the global adult population, ranging between 13.5% in Africa and 31.8% in the Middle East [5]. This surge in NAFLD rates is primarily due to the concurrent rise in the global rates of obesity, which is its primary cause [6].

There is currently no approved pharmacotherapy for NAFLD, and the only intervention proven to be significantly beneficial for NAFLD patients is weight loss [7]. Indeed, losing 5% to 10% of body weight improves abnormal liver tests, reduces liver fat, reduces inflammation and injury to liver cells, and may even reverse some fibrosis damage [8,9]. Unfortunately, losing the amount of weight required to improve NAFLD has been proven to be difficult, and maintaining that weight loss has been proven to be even more difficult [10,11]. As a result, and given the pressing burden of NAFLD, novel therapeutic approaches to improve NAFLD that are not dependent on weight loss represent a significant unmet medical need.

The Glucagon-Like Peptide-1 receptor (GLP-1R) agonists, which are already approved for the treatment of diabetes and obesity due to their weight loss-inducing effect [12,13,14,15], have recently been investigated to improve NAFLD in animals and humans, yielding promising results [16,17,18,19,20,21,22,23,24,25,26,27,28,29,30,31,32]. Consequently, these drugs are suggested as potential therapeutic options for treating and slowing the progression of NAFLD.

GLP-1 is a hormone secreted by the intestine’s L-cells [33]. It is best known for its “incretin effect” in restoring glucose homeostasis in people with diabetes by stimulating glucose-dependent insulin release, decreasing glucagon secretion, and promoting the proliferation of pancreatic beta cells [34]. However, it is now known that GLP-1 has a broader range of physiological effects in the body, including reducing endoplasmic reticulum stress, regulating autophagy, promoting metabolic reprogramming, activating anti-inflammatory signaling, changing gene expression, influencing neuroprotective pathways, slowing gastric emptying, and inhibiting satiety and food intake through actions on central nervous system centers [35,36]. This pleiotropic effect is due to the expression of the GLP-1 receptor by various organs, such as the pancreas, brain, kidney, gut, lung, heart, muscle, and liver [37].

Even though treatment with GLP-1R agonists improved NAFLD in vivo, particularly in terms of the liver fat content reduction, the mechanisms underlying this effect remain unknown. Specifically, it is unclear whether this effect results from the direct activation of the hepatic GLP-1R or the indirect effect of weight loss and, subsequent, insulin sensitivity, fat content reduction, and glycemic and lipid profile improvement. Gupta and colleagues [38] were the first to propose that the activation of the hepatic GLP-1R plays a direct role in reducing hepatic steatosis in vitro through modulation of the elements of the insulin signaling pathway. Recently, we and others [39,40,41] have shown that the GLP-1R agonist Exendin-4 (Ex-4) reduces the fat content in an in vitro cell model of steatosis by inhibiting hepatic lipogenesis through activation of β-catenin signaling and modulation of the expression of several lipogenesis genes. In addition, β-catenin might also mediate the effect of Ex-4 on ameliorating hepatic steatosis induced by a high fructose diet in rats [42].

A more in-depth understanding of the pathways through which GLP-1R agonists may alleviate steatosis is critical for clinical development as it may open new avenues for developing newer, safer, and more potent medications. This comprehension entails dissecting the molecular and metabolic changes in response to these medications. Given that NAFLD is a metabolic disease, understanding how GLP-1R agonists impact the structure of biomolecules, such as proteins, lipids, and carbohydrates, could provide valuable insights into the mechanisms underlying the protective effect of these agents on steatosis and NAFLD.

Fourier transform infrared (FTIR) spectroscopy is a fast and effective imaging method for obtaining important molecular and biochemical information from biological materials, without requiring any staining or special sample preparation. It can extract useful information on the specific structural changes in response to a disease state or a treatment, for example, by absorbing the IR wavelengths according to the structural and chemical bonds of the different molecules [43,44]. FTIR also allows for examination of the representative areas of samples, and because it uses infrared radiation as a source of excitation, the possibility of a destructive impact of the measurement on the sample is minimized [45]. FTIR has been successfully applied for studies of different tissues and cells, including the liver [46], brain [47], lungs [48], cervix [49], aorta [50], and human hepatocellular carcinoma cells (HepG2) [51,52]. As for the liver, FTIR was mostly used to examine liver cancer [46]; however, the technique is perfectly suited for analyzing the liver fat content [45].

The objective of the present study was to employ FTIR spectroscopy for investigating any structural changes in lipids and proteins that might occur in HepG2 cells made steatotic with oleic acid and then treated with the GLP-1R agonist Exendin-4. Unraveling these changes might provide more in-depth information about the mechanisms that underpin the protective effect of Ex-4 on steatosis at a molecular level.

## 2. Materials and Methods

### 2.1. HepG2 Culture

We obtained the human hepatoma HepG2 cell line (HB-8065, ATCC) from ATCC (Manassas, VA, USA) and kept it in Dulbecco’s Modified Eagle Medium (DMEM) (31966047, Gibco, Burlington, MA, USA) supplemented with 10% FBS (10500064, Gibco, MA, USA) and 1% penicillin/streptomycin (15070063, Gibco, MA, USA) at 37 °C and 5% CO_2_. We carried out all of the experiments with cells passaged no more than 25 times.

### 2.2. Preparation of Oleic Acid

We prepared the oleic acid (OA) solution as described in [53]. To summarize, we dissolved the powder OA (O-1008 Sigma-Aldrich, Germany) at a final concentration of 12 mM in phosphate-buffered saline (PBS; 137 mM NaCl, 10 mM phosphate, 2.7 mM KCl, and pH 7.4) containing 11% fatty acid free bovine serum albumin (FFA-BSA; 0215240110, MP Biomedicals, Santa Ana, CA, USA). The solution was then sonicated and shaken overnight at 37 °C with an OM10 Orbital Shaking Incubator (Ratek Instruments Pty, Ltd., Boronia, Australia). The OA solution was filtered with a 0.22 μM filter, aliquoted, and kept at 4 °C. We utilized a fresh aliquot for each experiment.

### 2.3. Induction of Steatosis and Treatment with Exendin-4

To create the steatosis cell model and treat it with Ex-4, we used the same procedure as in our recent publication [40]. In brief, we cultured HepG2 cells in six-well plates at a density of 4 × 10^5^ cells/well until 70% confluency, then starved them for 6 h in DMEM containing 1% fatty acid free FBS. Following starvation, we incubated the cells for 16 h at 37 °C in DMEM containing 400 μM and 1% fatty acid free FBS and then quantified steatosis. We used 1% fatty acid free FBS in all OA treatment experiments to ensure that OA was the single inducer in the medium and that OA does not react with components of FBS. Following steatosis induction, we washed the cells and incubated them for 3 h in fresh 1% FBS DMEM containing 400 μM OA solution in the absence or presence of 200 nM Ex-4 (E7144-1 MG, Tocris, Minneapolis, MN, USA). For each experiment, we used a fresh aliquot of EX-4. The optimal concentrations of OA and Ex-4 that we used were determined in our previous paper [40].

### 2.4. Quantification of Steatosis

To quantify steatosis, we used Oil Red O, which stains neutral triglycerides and lipids. We prepared an Oil Red O stock solution by dissolving 0.5 g of Oil Red O (Sigma O0625, Saint Louis, MO, USA) in 100 mL of 98% isopropyl alcohol. One hour before the staining, we prepared a working solution of 60% Oil Red O in PBS and filtered it through Whatman paper. We fixed the HepG2 cells with 4% paraformaldehyde for 15 min at room temperature, washed them thrice with PBS, and then incubated them with 2 mL of Oil Red O working solution in the dark for 1 h at room temperature. After three 5 min washes with PBS, microscopic images were taken to visualize the Oil Red O lipid droplet staining.

To quantify the lipid accumulation, we incubated the cells with 60% isopropanol for 30 s to extract the staining from the lipid droplets. A 96-well plate reader was used to immediately measure the fluorescence intensity of each sample at 490 nm excitation and 520 nm emission.

### 2.5. Sample Preparation for FTIR Analysis

Untreated, steatotic, and Ex-4-treated steatotic cells were detached with 3–5 min incubation in a solution containing 0.25% trypsin and 1 mM EDTA. They were then washed thrice with NaCl solution (0.9%) to remove the trypsin and culture medium before being resuspended in 1 mL of 4% paraformaldehyde and shaken at room temperature for 30 min. The 4% PFA was washed away with sterile water. Approximately 1 million cells from each condition were resuspended in 0.1 mL water, mounted on BaF_2_ discs (25 × 4 mm) (Thermo Fisher, Scientific, Waltham, MA, USA), and allowed to dry overnight in a special cabinet. We manipulated cells from the three conditions similarly, allowing for an accurate comparison of the experimental spectroscopic data.

### 2.6. FTIR Data Collection

We performed the FTIR measurements as described previously [54]. We used a Nicolet iS50 FTIR spectrophotometer in absorption mode within the range of 4000–900 cm^−1^, with 64 scans per spectrum and 4 cm^−1^ spectral resolution. Background single beam spectra were measured on a substrate without biological cells by co-adding 128 scans. We collected the FTIR spectra from three different cell cultures for each condition, and 40 spectra from different locations were acquired from each sample. We performed data collection using the Thermo Scientific™ OMNIC™ Series Software, version 9. The spectra were acquired at the cell layer sample away from the border between the cell layer and the substrate to avoid resonance Mie scattering (commonly present in biological samples).

### 2.7. FTIR Data Processing and Analysis

The acquired FTIR spectra were processed using MATLAB (version 2015a, MathWorks, Natick, MA, USA). Each spectrum was normalized by dividing it by the area under the curve of the full spectrum. The normalized spectra were baseline corrected to remove any systematic noise in the spectra. We calculated the relative amount of different protein contents using curve fitting [55]. To quantify the amounts of different protein types, we fitted a linear mixed model with Gaussian peaks centered on the wavenumbers of the protein types. The curve fitting was performed over the spectral range of 1700–1600 cm^−1^ to identify α-helix, random coil, β-turn, and β-sheet (parallel and antiparallel). To compare the integrated intensities of different peaks and between different treatments, we used ANOVA followed by Turkey’s post hoc test. Statistical significance was considered at *p* < 0.05. The spectral band assignments for different biochemical contents are available in [56].

### 2.8. Chemometric Analysis

The infrared spectra were converted from Nicolet system format to comma-separated values (CSV) to create suitable data sets for the analysis. We performed chemometric analyses, including principal component analysis (PCA) and hierarchical cluster analysis (HCA), to investigate the significant difference between the spectra acquired from the different treatment conditions. To obtain the PCA plots, we used the pracma package in R (https://cran.r-project.org/web/packages/pracma/index.html (accessed on 21 February 2022), while for the hierarchical cluster, we used the A2Rplotfunction from the A2R package in R [57]. PCA allows for noise reduction and captures subtle differences within the spectral collection [58]. An IR spectrum composed of *p* = 6224 biologically meaningful wavenumbers between 4000 cm^−1^ and 1000 cm^−1^ was initially used for the PCA analysis. In our experiments, we selected the top *q* = 3 PCs as they captured approximately 100% of the variance in the data. Next, we used these eigenvectors to generate the loading vectors, which effectively captured the wavelengths corresponding to the variations between the spectra of different treatment experiments [59].

The primary results obtained from the PCA were the score and loading plots. The score plots represent the spectra of these samples in a *q*-dimensional space of PCs. The loadings plot shows which wavelengths are responsible for the data set for the most significant degree of separation within this spectral collection. The 3D score plot of the PCs that explain the majority of the variance in the dataset enabled the spectra to be grouped according to the chemical information they contain [58]. Hierarchical cluster analysis was applied using the A2R package in R to compare the control and treated samples based on their PCA projections in the 3D score space [60]. HCA was used to group the spectra that displayed the same degree of similarity by using Ward’s algorithm to calculate the Euclidean distance between all of the data sets. The result was visualized as a dendrogram, and the sample grouping was presented as samples made up of color clusters based on the heterogeneity scale [60].

## 3. Results

### 3.1. Ex-4 Significantly Reduces OA-Induced Lipid Accumulation in HepG2

We have previously shown that the incubation of HepG2 cells with 400 µM OA after 6 h of starvation induces steatosis, as reflected by the presence of lipid droplets [39,40]. We have further shown that treatment of the steatotic HepG2 cells with 200 nM Ex-4 for 3 h resulted in a significant reduction in steatosis [40], indicating the protective effect of this drug on steatosis. This study quantified lipid accumulation using ORO staining. As shown in Figure 1, 400 µM OA caused a significant increase in lipid content, which was significantly reversed upon treatment with 200 nM Ex-4 for 3 h.

### 3.2. FTIR Spectroscopic Profiling of Total Lipids and Proteins

The FTIR band assignments in this study are based on the specific spectral bands, as defined in Appendix A [61]. To examine the potential changes in the total lipids and proteins in response to different treatments, we performed an FTIR analysis in the 4000–1000 cm^−1^ frequency range (Figure 2A).

The FTIR spectra of the three treatment conditions show the characteristic lipid bands between 3050 and 2800 cm^−1^ and between 1800 and 1700 cm^−1^ (Figure 2B,C). The overlapping hidden peaks were identified and reconstructed by performing peak fitting on the lipid region of the spectra (Figure 3). The 3050–2080 cm^−1^ region essentially contains C-H stretching bands from various vibrational modes, including asymmetrical stretching (V_as_) (CH_2_) near 2920 cm^−1^, symmetrical stretching (V_s_) (CH_2_) near 2850 cm^−1^, V_as_(CH_3_) near 2960 cm^−1^, V_s_(CH_3_) near 2870 cm^−1^, and the olefinic =CH (V_as_ (CH=CH) content at 3027–3000 cm^−1^ (Figure 2B). Another key peak for the lipid analysis occurs in the lower wavenumber region at 1740 cm^−1^ and is assigned to C=O stretches of the ester functional groups from lipid triglycerides and fatty acids, reflecting the total lipids in the cell (Figure 2C).

The samples from the three treatment conditions have absorption bands at similar wavenumbers, but with different intensities, denoting the same components but with different concentrations. For instance, the results show that the V_as_(CH_2_): V_as_(CH_3_) ratio is significantly lower in steatotic cells than in untreated cells (1.53 ± 0.19 versus 1.65 ± 0.21; *p* < 0.01) (Figure 4A), suggesting a lower concentration of long-chain fatty acids in steatotic cells. Interestingly, treatment with Ex-4 significantly reverses the V_as_(CH_2_): V_as_(CH_3_) ratio to similar levels as the untreated cells (Figure 3A). The V_s_(CH_2_):V_s_(CH_3_) ratio is also significantly lower in steatotic cells compared with the untreated cells (0.97 ± 0.08 versus 1.35 ± 0.05; *p* < 0.001), and again Ex-4 treatment significantly reverses this ratio (1.14 ± 0.12, *p* < 0.001), although it remains significantly lower than in the untreated cells (*p* < 0.001) (Figure 4B). These observations suggest the abundance of short-chain fatty acids in untreated cells compared with steatotic and Ex-4-treated cells.

Additionally, the V_as_(CH_2_)/V_as_(CH=CH) ratio, which reflects the unsaturation index, is significantly higher in steatotic and Ex-4-treated cells than in the untreated cells (Figure 4C), indicating a higher degree of unsaturation in the fatty acids of the steatotic and Ex-4-treated cells.

Finally, we observed a significant increase in the phospholipid band (C=O) at 1740 cm^−1^ in the steatotic cells compared with untreated cells (*p* < 0.001), indicating, as expected, lipid accumulation. Interestingly, upon treatment with Ex-4, the phospholipid band (C=O) at 1740 cm^−1^ is significantly decreased compared with the steatotic cells, suggesting a decrease in lipid accumulation in response to Ex-4 treatment (Figure 4D).

### 3.3. FTIR Spectroscopic Profiling of Total Proteins

Next, we explored the secondary structure of the cell proteins under the three treatment conditions. We investigated the IR absorption in the frequency range of 1450–1750 cm^−1^ (Figure 3), which essentially contains stretching bands characteristic of proteins.

The amide I band is composed of many contributions assigned to β-sheet within 1635–1610 cm^−1^ (yellow line in Figure 4), random coil within 1645–1630 cm^−1^ (brown line in Figure 3), α-helix within 1660–1650 cm^−1^ (red line in Figure 4), antiparallel β-sheet (green line in Figure 3), and β-turn (blue line in Figure 3) within 1695–1665 cm^−1^ (Appendix A). The relative amount of these protein structures was quantified from the peak fitting of the original absorbance spectra.

A comparison of the integrated intensities of the different spectral bands revealed a significant increase in the amide I, β-turn, α-helix, β-sheet, and amide II contents in the steatotic cells compared with the untreated cells (Figure 5A). Interestingly, this increase is significantly reversed after treatment with Ex-4 (Figure 5A), although it remains significantly higher than the untreated cells for α-helix, β-sheet, and amide II. In contrast, the random-coil content is significantly reduced in the steatotic cells compared with the untreated cells. Again, this reduction is reversed by Ex-4 treatment, although it remains significantly higher than in the untreated cells.

The significant increase in the amide I band intensity and the absence of any significant band shift in steatotic cells compared with both the untreated and Ex-4-treated cells (Figure 5A, *p* < 0.001) is suggestive of a rise in protein content, probably because of the increased demand of enzymes needed for lipid accumulation handling.

The results also showed that the amide I components overlap, and the band ratio of the α-helical/amide I composition was significantly higher in the steatotic cells than in the untreated cells (0.68 ± 0.03 versus 0.55 ± 0.04, *p* < 0.001). After treatment with Ex-4, this ratio decreased significantly (0.68 ± 0.03 versus 0.60 ± 0.07, *p* < 001), but it remained significantly higher than in the untreated cells (0.60 ± 0.07 versus 0.55 ± 0.04, *p* < 0.001) (Figure 5B). We observed a similar trend for the ratio of the β-sheet/amide I composition between the untreated and steatotic cells, but the treatment with Ex-4 did not significantly reverse the effect of OA (Figure 5B). In contrast, the ratio of the random coil/amide I composition decreased significantly after the OA treatment (0.28 ± 0.04 vs 0.16 ± 0.03, *p* < 0.01), but then increased significantly after Ex-4 treatment (0.16 ± 0.03 versus 0.22 ± 0.06, *p* < 0.001), although it remained significantly lower than the untreated cells (0.22 ± 0.06 versus 0.28 ± 0.04, *p* < 0.001). The results also showed that the band ratio β-sheet/α-helical was significantly increased in the steatotic cells versus the untreated cells (0.48 ± 0.05 versus 0.43 ± 0.09, *p* < 0.01) and that Ex-4 did not reverse this increase (0.48 ± 0.05 versus 0.48 ± 0.05, *p* > 0.99) (Figure 5B). These results indicate that the untreated control cells retained their native functional α-helical structure.

### 3.4. Chemometric Data Analysis

In addition to the FTIR data, we performed principal component analysis (PCA) on the baselined collected spectra in the 3100–2750 cm^−1^ range (Figure 6). The principal component (PC) scatterplot presented in Figure 6A shows a distinct separation between the untreated, steatotic, and Ex-4-treated cells. The first principal component (PC1) accounted for 84% of the total variance in the data, while PC2 and PC3 accounted for 11% and 4.5%, respectively. The relative contribution of the first three PCs to the cumulative variance is depicted in Figure 6B. All of the other principal components would be associated with the noise in the spectral collection, as the three first PCs explain approximately 99.22% of the variance in the data.

As depicted in Figure 6C, the loading corresponding to PC1 can capture significant variations between wavenumbers 3020 and 2840 cm^−1^. The loading vector corresponding to PC2 can capture larger variations between wavenumbers 3005 and 2975 cm^−1^, the region where PC1 can capture the relatively lower variance.

Figure 6D shows that PC1 has a high R^2^ and can explain most of the regions of the spectra between 3100 and 2750 cm^−1^ considered during chemometric analysis. PC2 contributes to R^2^ in the wavenumbers between 3005 and 2975 cm^−1^ and between 2845 and 2835 cm^−1^. PC3, on the other hand, has a small contribution to R^2^ in the wavenumbers between 3020 and 3017 cm^−1^, and between 2840 and 2836 cm^−1^.

Using the PCA results, we generated a hierarchical dendrogram on all of the spectral samples between the wavenumbers 3100 and 2750 cm^−1^ belonging to the untreated, steatotic, and Ex-4-treated samples. For each cluster, we generated the representative low-dimensional embedding for that cluster by taking the mean from all of the points in Figure 6A that belonged to either the untreated, steatotic, or Ex-4-treated samples, respectively. We then generated a dendrogram using the “A2Rplot” function from the A2R package in R. The resulting dendrogram is depicted in Figure 6A.

Finally, we created a PCA biplot, which could explain the differences between the different treatment samples by revealing which spectral regions are more connected to which sample type. As shown in Figure 6F, most of the spectra in the 2767−2796 cm^−1^ region are associated with the untreated samples, while the spectra in the 2887–2934 cm^−1^ region are associated with steatotic cells, and most of the spectra in the 2806–2995 cm^−1^ region are associated with the Ex-4 treated samples, indicating differences in the lipid content between the three treatment conditions.

To investigate the cellular protein’s conformation changes in depth, we restricted the PCA analysis to the 1750–1450 cm^−1^ region to explore whether the spectra from the different treatment conditions could be distinguished by focusing only on the protein region. As with the lipids, the first three principal components (PCs) captured approximately 98% of the total variance in this spectral region (Figure 7A,B). All of the other principal components would be associated with the noise in the spectral collection. The loading plots in Figure 7C show which wavenumbers in the data set are responsible for the maximum degree of separation inside this spectral region. The loading corresponding to PC1 is positively correlated with wavenumbers between 1750 and 1645 cm^−1^, but negatively correlated with wavenumbers between 1645 and 1480 cm^−1^, and can capture significant variations between these wavenumbers. PC1, again, helps distinguish the spectral samples belonging to the three treatment conditions.

Figure 7C indicates that PC1 can explain most of the regions of the spectra considered during the chemometric analysis. In particular, PC1 has a high R^2^ and can explain the regions between the wavenumbers 1750–1660 cm^−1^, 1635–1590 cm^−1^, and 1545–1475 cm^−1^ well. PC2 contributes to R^2^ in the wavenumbers between 1700–1600 cm^−1^ and 1600–1530 cm^−1^. Finally, PC3 has a small contribution in R^2^ in the wavenumbers between 1670–1575 cm^−1^. Furthermore, we performed hierarchical clustering of all of the spectra from the protein region belonging to the three treatment conditions (Figure 7E). For each cluster, we generated the representative low-dimensional embedding for that cluster by taking the mean from all of the points in Figure 7A that belong to either the untreated, steatotic, or Ex-4-treated samples. As shown in Figure 7E, the samples from each treatment condition cluster together and are distinguished from one another. As for the lipids, we generated a PCA biplot, as shown in Figure 7F. Most of the 1511–1514 cm^−1^ region spectra are associated with the untreated samples, while the spectra in the 1551–1610 cm^−1^ and 1656–1723 cm^−1^ regions are associated with the Ex-4-treated and steatotic samples, respectively.

## 4. Discussion

In the present study, we combined FTIR and chemometric analyses to biochemically characterize and differentiate between untreated, steatotic, and steatotic HepG2 cells treated with the GLP-1R agonist Ex-4. To the best of our knowledge, this is the first study to use FTIR to investigate the biochemical changes induced by steatosis and treatment with a GLP-1R agonist. One of the key benefits of using FTIR spectroscopy is the possibility of using intact cells and analyzing the entire biochemical signature, including the proteins, lipids, nucleic acids, and carbohydrates [64].

The FTIR spectral analysis indicated, as expected, that Ex-4 reduces the lipid accumulation induced by OA and that steatosis causes alteration in the lipids and proteins, most likely due to lipotoxicity-induced oxidative stress. Several studies have previously demonstrated that OA induces lipid accumulation in HepG2, which in turn leads to lipotoxicit, as well as oxidative and ER stress [65,66,67,68,69,70,71,72,73]. We found significant differences in the lipid acyl (CH_2_), the methyl group (CH_3_), the lipid ester (C=O), and the olefinic CH=CH contents between the three treatment conditions. A decrease in the lipid acyl (CH_2_)/methyl group (CH_3_) ratio in the steatotic cells compared with the untreated cells, along with an increase in the phospholipid content (C=O), could indicate the accumulation of short-chain fatty acids in response to OA. This observation might also indicate the degradation of the long-chain fatty acids into short-chain fragments due to lipid peroxidation. Hepatic steatosis has indeed been shown to be associated with lipid peroxidation and hepatic fibrosis in a variety of liver diseases, including NAFLD [74]. Our results also show an increase in the olefinic (CH=CH) double bond at 3020–2993 cm^−1^ content in the steatotic cells, suggesting a higher degree of unsaturation. Interestingly, compared with the effect observed between the untreated cells and OA-treated cells, Ex-4 treatment resulted in a small but significant reversal of the lipid acyl (CH_2_)/methyl group (CH_3_) ratio and a reduction in the phospholipid content (C=O) compared with the steatotic cells. This finding supports our and others’ findings that Ex-4 reduces lipid accumulation in steatotic HepG2 cells [39,40,41]. It has indeed been reported that Ex-4 may inhibit lipotoxicity-induced oxidative stress in pancreatic β-cells by inhibiting the activation of the TLR4/NF-κB signaling pathway [75]. Furthermore, Erdogdu and coworkers have shown that Ex-4, through a GLP-1R-dependent mechanism, protects endothelial cells from lipoapoptosis associated with lipotoxicity [76]. In our case, Ex-4 did not affect the degree of FA unsaturation.

Protein changes caused by oxidative stress can result in losing a protein’s secondary or tertiary structure, thus affecting its function, stability, and solubility. Furthermore, regulated post-translational protein changes that are critical for the stability of the cellular proteome are catalyzed by enzymes that can be harmed by oxidative stress [77].

The curve fitting of the amide I band at 1700–1600 cm^−1^ was used to investigate the level of protein aggregation. We observed a significant increase in the amide I band in the steatotic cells relative to the untreated cells. We also observed a significant increase in the band absorption of β-turns (≈1680), α-helical (≈1650–1660 cm^−1^), and β-sheets (≈1625–1635 cm^−1^) in the steatotic cells relative to the untreated cells. These findings are suggestive of potential protein aggregation and/or protein malfunction. Interestingly, these increases were all significantly reversed upon Ex-4 treatment, although the effect seemed smaller compared with the effect between the control and OA-treated cells. The mechanisms underlying this reversal remain to be identified.

The combined results from the chemometric, principal component, and hierarchical cluster analyses revealed a clear separation between untreated, steatotic, and Ex-4-treated samples, suggesting that the samples experienced significant molecular changes in their biochemical makeup. The PCA score plots for both the lipid and protein regions showed that the untreated and EX-4-treated samples, while still separated, were clustered close to each other, far from the steatotic cells. This observation suggests that treating the steatotic cells with Ex-4 does, at least partially, reverse the biochemical and structural changes.

Despite the high sensitivity of FTIR, our study is not without limitations. Specifically, we did not investigate the biochemical changes in response to other fatty acids such as palmitic acid or a mixture of palmitic acid and oleic acid. In addition, although FTIR can detect subtle structural changes in response to OA, it is unable to tell how important these are for NAFLD development or progression. In future studies, we plan to use FTIR to investigate the biochemical changes in response to other treatments and to correlate the structural changes with different stages of NAFLD using liver biopsies from humans.

## 5. Conclusions

Our FTIR and chemometric analyses indicate that the Ex-4 treatment of steatotic HepG2 cells reduces OA-induced lipid accumulation and restores the lipid and protein biochemical alterations caused by lipotoxicity-induced oxidative stress. Furthermore, our findings demonstrate that FTIR spectroscopy is a fast, non-destructive and refined technique for detecting biochemical and subcellular changes. Regardless of the biochemical explanation for the variations in the FTIR spectra we obtained, our data show that optical spectroscopy can be a good tool to study the impact of different agents on the cell’s biochemical components, and the observed structural changes are correlated to the cellular functions or phenotype. This opens the possibility of using optical techniques to monitor cellular organic components in situ, providing a valuable tool for diagnosing pathologies and monitoring changes in the tissues during therapeutic treatment. In the case of NAFLD, for example, FTIR could be used on liver biopsies to investigate general changes in the cell biochemistry, e.g., during different stages of NAFLD, and to monitor the response to treatments. It can also be used to monitor carcinogenesis, necrosis, or apoptosis. Another advantage of FTIR is the low running cost compared with biochemical assays.

## Figures and Tables

**Figure 1 biomedicines-10-02652-f001:**
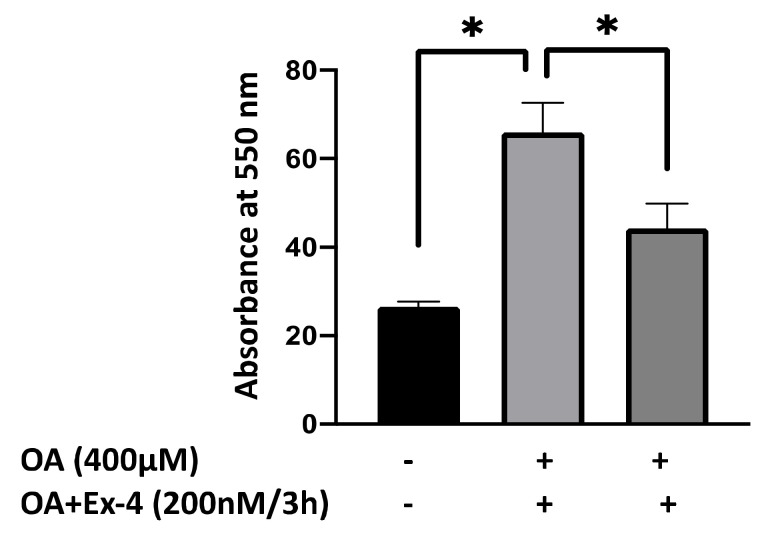
Oil Red O staining of lipid droplets and quantification of lipid accumulation. HepG2 cells treated with 400 µM OA show increased staining for Oil Red O, indicating lipid accumulation intracellularly. The quantification of lipid accumulation in HepG2 cells treated with OA for 16 h, followed by treatment with 200 nM Exendin-4 for 3 h. The means were compared using Student’s *t*-test. The values are mean ± SEM. Statistical significance was considered at *p* < 0.05. The results are the average of four experiments for each condition. * indicates *p* < 0.05.

**Figure 2 biomedicines-10-02652-f002:**
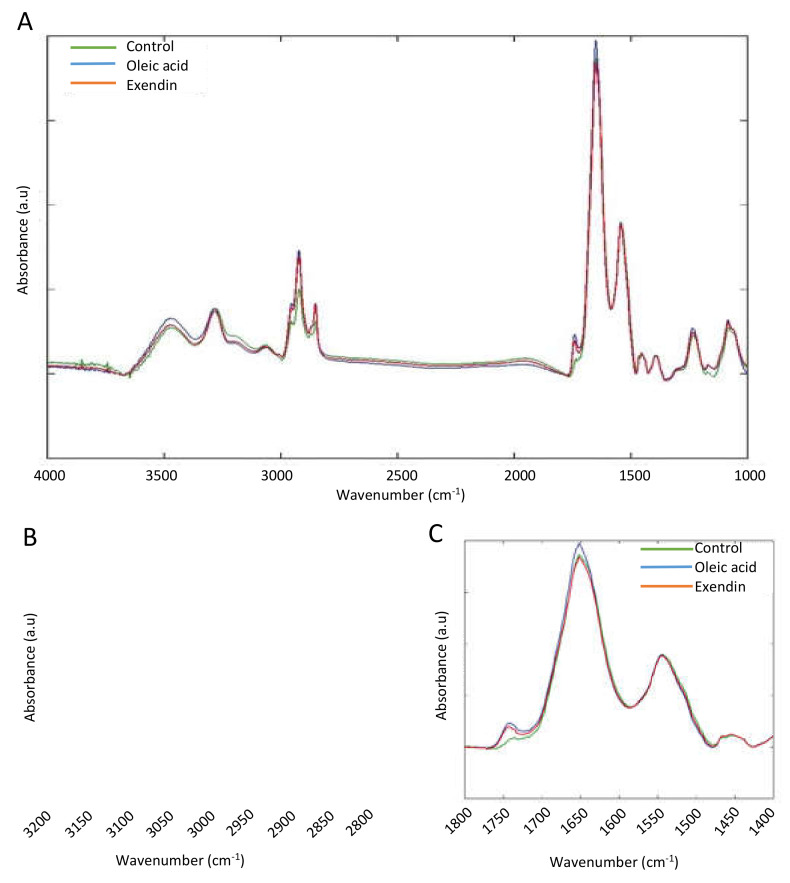
(**A**). Average full FTIR spectra of untreated HepG2 cells (green line), steatotic HepG2 cells (blue line), and steatotic HepG2 cells treated with Exendin-4 (red line). (**B**) Close-up of the FTIR spectra of the lipid region in the frequency region of 3200–2800 cm^−1^. (**C**) Close-up of the protein region in the frequency region of 1800–1400 cm^−1^. Each of the three spectra is the average of all of the spectra collected from three different experiments for each condition.

**Figure 3 biomedicines-10-02652-f003:**
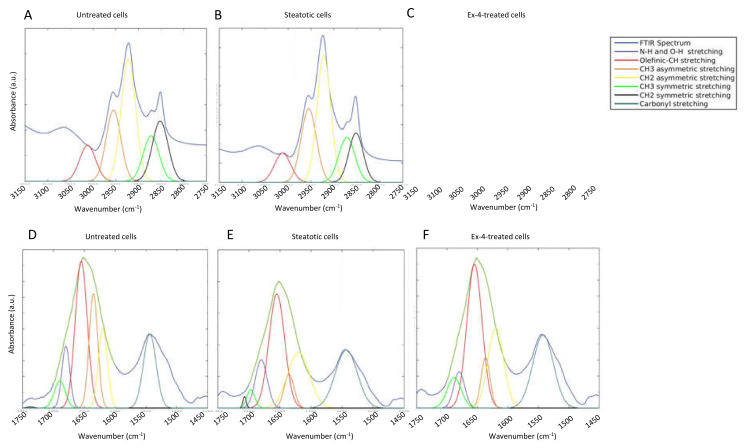
Curve fitting of the FTIR spectra. (**A**–**C**) Curve fitting of the lipid region in the mean FTIR spectra of the untreated cells, steatotic cells, and steatotic cells treated with Exendin-4. (**D**–**F**) Curve fitting of amide I and II regions in the mean FTIR spectra of the untreated cells, steatotic cells, and steatotic cells treated with Exendin-4. The overlapping hidden peaks were identified and reconstructed by performing peak fitting on the spectra. The amide I band (1700–1600 cm^−1^) arises mainly from the backbone C=O stretching vibrations (~80%) of the amide groups coupled with little in-plane NH bending (< 20%) [62], while the amide II band results from backbone N-H bending and C-N stretching vibrations in the spectral range of 1580–1510 cm^−1^ [63]. The most critical and frequent band used to explain the secondary structure of polypeptides is the amide I band, as the amide II and amide III bands have multiple assignment contributions [54].

**Figure 4 biomedicines-10-02652-f004:**
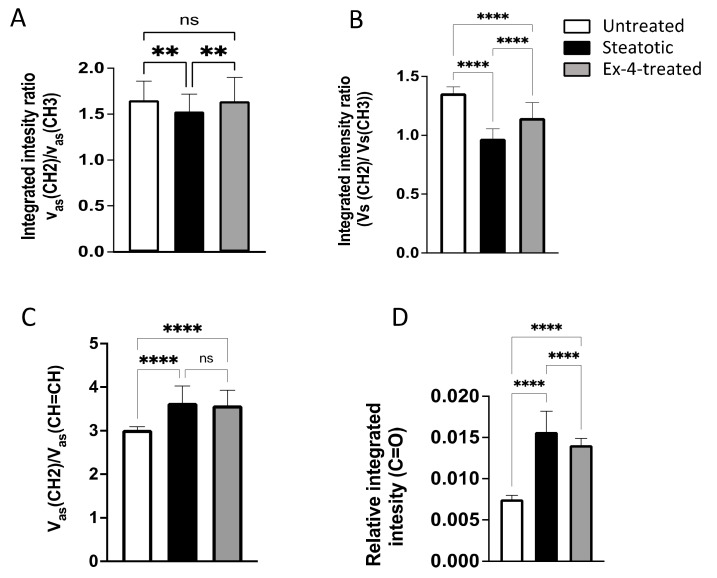
Comparison of several representative spectral parameters between untreated, steatotic, and Ex-4-treated HepG2 cells. (**A**) V_as_(CH_2_):V_as_(CH_3_) ratio. (**B**) V_s_(CH_2_):V_s_(CH_3_) ratio. The (**C**) V_as_(CH_2_):V_as_(CH=CH) ratio. (**D**) The phospholipid band (C=O) at 1740 cm^−1^. All the values are expressed as the mean ± SE. The FTIR spectra analyzed were 70, 80, and 80 for the untreated, steatotic, and Ex-4-treated HepG2 cells, respectively. Each treatment condition was repeated three times. The comparison between the different treatments was performed using ANOVA, followed by Turkey’s post hoc test. ** and **** indicate *p* < 0.01 and *p* < 0.0001, respectively. ns: non-significant.

**Figure 5 biomedicines-10-02652-f005:**
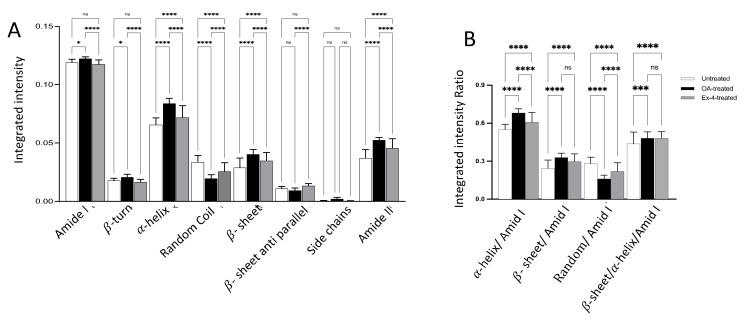
Comparison of several representative spectral parameters in the protein region between the untreated, steatotic, and Ex-4-treated HepG2 cells. (**A**) Relative proportion of the secondary structures of polypeptides in the protein region. (**B**) Integrated intensity ratios of the different secondary structures of polypeptides. All of the values are expressed as the mean ± SE. The FTIR spectra analyzed were 70, 80, and 80 for the untreated, steatotic, and Ex-4-treated HepG2 cells, respectively. Each treatment condition was repeated three times. The comparison between different treatments was performed using ANOVA followed by Turkey’s post hoc test. *, ***, and **** indicate *p* < 0.05, *p* < 0.001, and *p* < 0.0001, respectively. ns indicates non-significant.

**Figure 6 biomedicines-10-02652-f006:**
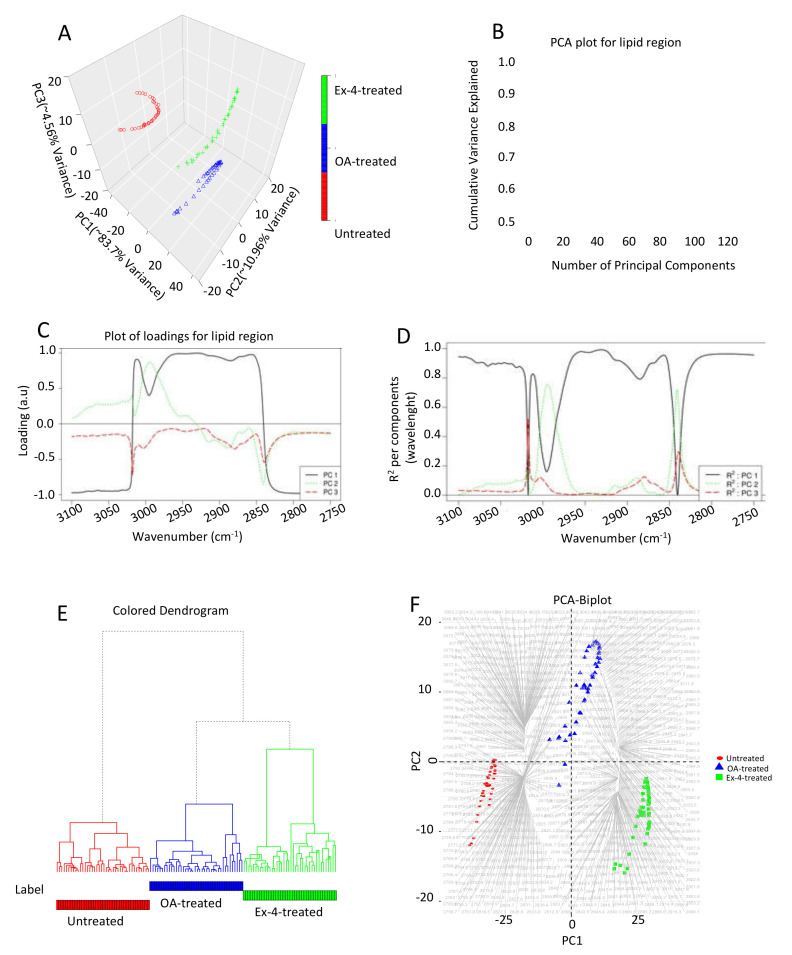
Chemometric analysis of FTIR experimental data on the lipid region (3100–2750 cm^−1^). (**A**) PCA score plot. (**B**) Amount of variance captured by PCs. (**C**) Loading plots for the spectral data. (**D**) Squared correlation for the first three PCs for the spectral collection. (**E**) Hierarchical clustering of the spectra for the different experimental samples. (**F**) PCA biplot explaining which wavenumbers are associated with which treatment samples. Each treatment condition was repeated three times. The FTIR spectra analyzed were 70, 80, and 80 for the untreated, steatotic, and Ex-4-treated HepG2 cells, respectively.

**Figure 7 biomedicines-10-02652-f007:**
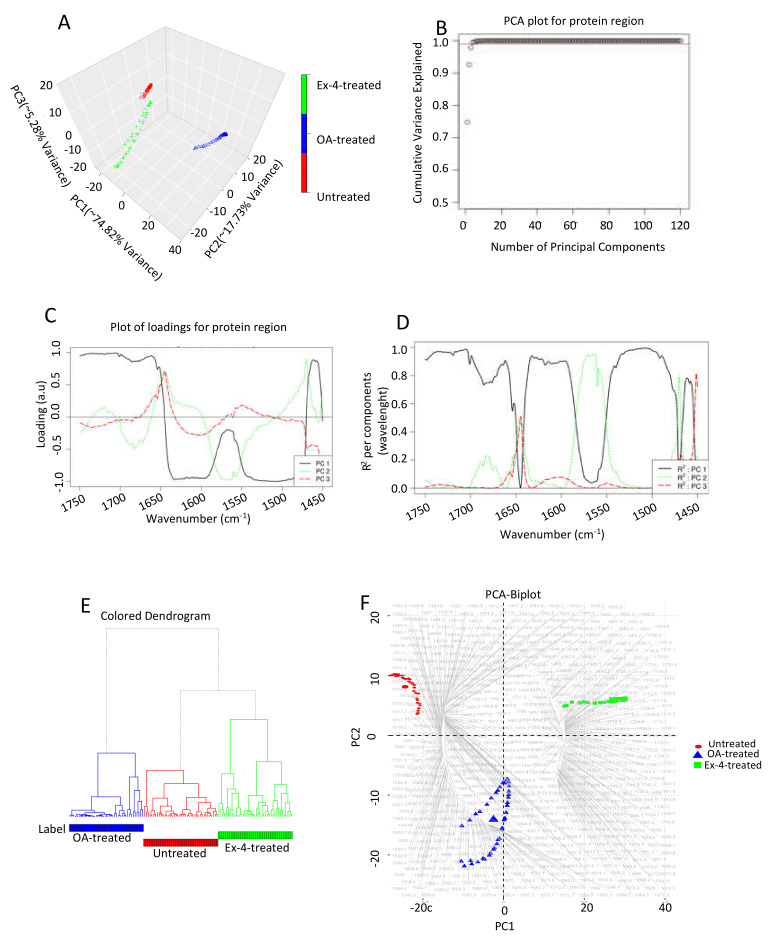
Chemometric analysis of the FTIR experimental data on the protein region (1750–1450 cm^−1^). (**A**) PCA score plot. (**B**) Amount of variance captured by PCs. (**C**) Loading plots for the spectral data. (**D**) Squared correlation for the first three PCs for the spectral collection. (**E**) Hierarchical clustering of the spectra for the different experimental samples. (**F**) PCA biplot explaining which wavenumbers are associated with which treatment samples. Each treatment condition was repeated three times. The FTIR spectra analyzed were 70, 80, and 80 for the untreated, steatotic, and Ex-4-treated HepG2 cells, respectively.

## Data Availability

Not applicable.

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
