# Peer review of "Investigation of the Effect of Exendin-4 on Oleic Acid-Induced Steatosis in HepG2 Cells Using Fourier Transform Infrared Spectroscopy"

_biomedicines, 2022, doi:10.3390/biomedicines10102652_

Round 1

Reviewer 1 Report

This study performed FTIR and chemometric analyses to demonstrate that Ex-4 treatment of steatotic HepG2 cells reduces OA-induced lipid accumulation and restores the lipid and protein biochemical alterations caused by OA treatment. Although FTIR analysis provided some new profile on the effect of EX-4 treatment, I have the following concerns that the authors need to address:

1. There are some evidence indicating that unsaturated fatty acids including OA are protective other than toxic compared with saturated fatty acids such as palmitate. Although high concentration of OA (e.g., 0.4 mM) can induce steatosis, it may not induce oxidative stress. The OA-induced steatosis may not be a good model for studying NAFLD, which is the context where this study is carried out. This paper did not cite any references or provide experimental data for the lipotoxicity of OA (e.g., oxidative stress, endoplasmic reticulum stress).

2. Although FTIR analysis could provide information on the structural changes of lipids and proteins caused by OA treatment, these structural changes may be due to the fatty acid compositional changes resulted from OA uptake or OA-induced effects, and the fatty acid compositional changes may not be toxic or harmful to the cells. In other words, FTIR analysis only provides information on the structural changes of lipids and proteins, but does not tell if the changes are harmful to the cells. This limitation restricts the usefulness of FTIR in NFALD study. In addition, different fatty acids may cause different structural changes of lipids and proteins, and thus this study may use another fatty acid or more fatty acids to do FTIR analysis, which may provide additional information to evaluate the usefulness of FTIR in NAFLD study.

3. Some claims or conclusions do not have enough data or evidence to support, and some explanations are arbitrary. For example, Line 29, the authors claim OA-induced lipotoxicity, but there are no data indicating the lipotoxicity induced by OA in this study; Line 31, the authors claim that ‘the lipid and protein biochemical alterations caused by lipotoxicity-induced oxidative stress’, but there no data or evidence to support this claim; similar problems are also present in the discussion. Please double check and revise.

4. How was the concentration of Ex-4 determined for in vitro study? Please provide the rationale.

5. Minor editing is needed. For example, some words should not be capitalized in title and subtitles, some abbreviations should be defined at their first use, some Greek letters are not correctly presented, cm-1 should be cm-1, the unit ml should be mL, there should be a space between the values and the units, there are redundant spaces between words somewhere in the text, the word represent should not be italic (Line 56), etc.

6. Line 114, it is not correct to say that kept in DMEM, it should be kept in DMEM supplemented with 10% FBS and 1% penicillin/streptomycin or kept in the complete medium and the complete medium consisted of DMEM, 10% FBS and 1% penicillin/streptomycin.

7. Line 126, 0.22 m should be 0.22 μM.

8. Line 131, does the word starved mean serum-starved? Why did the researchers do starvation for in vitro study?

9. Line 132-133, was the old media discarded before adding media containing OA treatment? For OA treatment, did the media still contain no serum? But based on the description that the cells were starved before OA treatment, the media containing OA should have serum. The authors should provide more details. The same question for the media mentioned in Line 135.

10. The authors may discuss the possibility of using FTIR in vivo study of NAFLD.

11. The effect of EX-1 treatment on the structural changes of lipids and proteins seemed to be very limited.

12. The researchers may provide more information on the OA-induced steatosis model, for example, the triacylglyceride content in the cell, the oxidative stress level, etc.

13. The authors claim FTIR analysis is superior to time-consuming biochemical assays. Please remove it, because there was no comparion of FTIR with other biochemical methods.

Author Response

Dear Reviewer,

Kindly find attached the answers of your comments. 

Regards,

Reviewer 2 Report

biomedicines-1756644

The authors have employed FTIR spectroscopy for investigating any structural changes of lipids and protein structures that might occur in HepG2 cells made steatotic with oleic acid and then treated with the GLP-1R agonist Exendin-4. Unraveling these changes might provide more in-depth information about the mechanisms that underpin the protective effect of Ex-4 on steatosis at the molecular level. However, some issues and concerns about the generalizability and usefulness of the model should be clarified or need to have further revision.

1.     The authors presented the quantification of lipid accumulation and showed the average of results from 4 cell experiments for each condition. Mann Whitney-U test was performed on all data to identify significant differences (p < 0.05 with a 95% confidence interval). I am curious about the numbers of the experiments and the selection of the statistic method. (1) Why do the authors not perform 3, 5 or 10 experiments for each conditions to obtain the results in Figure 1 ? (2) The results were collected from triplicate experiments in Figure 3 and Figure 5, and make a significant differentiation between three experiments. However, the average of each condition might not have observational difference and fold change. Do the authors use other statistic results to analyze the difference? Please provide the comparison (at least 3 methods) to have the confidence for the results with slight change of average ratio (i.e. Fig 3 and Fig 5).

2.     In Figure 6F and 7F, the colors of experimental conditions are different with its figureA&B. Please modify the color for the consistence of data presentation.

3.     How many samples are presented in Figure 6E & 7E? Please provide more interpretations for these figures.

4.     In conclusion, the author mentioned that “FTIR spectroscopy is a fast non-destructive and refined technique for detecting biochemical and subcellular changes whose identification typically requires time-consuming biochemical assays that may be insufficiently sensitive to detect minute structural, molecular, and sub-cellular changes.” However, FTIR spectroscopy can not identify the differential target protein or biochemical molecule. I suggest the authors can revise this sentence and describe more advantages and disadvantages of FTIR technique in this study (or related studies).

Minor revision:

1.     The resolution of all spectra including the number of axis are not good to read. Please provide high resolution of spectral picture or redraw all of the spectra, such as Figure 2, 4, 6, and 7..

2.     Please correct all of the “micro” for concentration in main text and figures. Do not use “u” or strange symbol.

3.     Page 4, line 157: The PFA 4% should be “The 4% PFA”.

4.     Page 7, line 260: CH2 should be revised as CH2.

5.     All of the text size in figures should be visualized and also should have the modest and consistent size/format in the same figure.

6.     Please check the spelling and capital/lowercase in the main text.

Author Response

Dear Reviewer,

Please find attached the responses to your comments.

Regards,

Round 2

Reviewer 1 Report

Thanks for the authors responses to my critiques. Most of the critiques have been addressed. However, I still have the following concerns in regard to the revisions that the authors made to the manuscript.

1. In the authors response to my comment: We value the reviewer's efforts and commendable remarks. We concur with the reviewer that various fatty acids may result in different structural alterations in proteins and lipids. Our primary goal was to investigate if we could use FTIR to detect Exendin-4's impact on Oleic acid-induced steatosis in HepG2 cells. We did not investigate whether or not the structural alterations to lipids and proteins are detrimental to cellular health. In our tests, we only exposed the cells to OA for 16 hours, and we expect longer OA treatments to lead to cellular death. Based on the outcomes shown in this study with Oleic acid, we intend to do FTIR analysis with various fatty acids in subsequent investigations. Also, our results indicate that FTIR could be used to assist in diagnosing NAFLD stages based on changes in biochemical components in liver biopsies. FTIR can also be used to monitor the response to various NAFLD treatments. I suggest that in the discussion section, the authors should mention the limitations of FTIR or what need to be further investigated in the future. The pathological interpretation of OA-induced compositional changes detected by FTIR and whether the compositional changes are varied with different fatty acids both are important to NAFLD study.

2. In the authors response to my comment: We appreciate the feedback and suggestions. We did not look at the OA-induced lipotoxicity and oxidative stress, but, as we said in our response to the first question, several studies did. In light of our response to the first query above, we have amended the discussion to include references that indicate that OA induces oxidative stress in HepG2 cells. Please see line 434 and 448 in the clean copy. I think the authors should remove such statements in the manuscript because there were no data showing EX-4 ameliorated OA-induced lipotoxicity although previous studies report OA-induced lipotoxicity.

3. In the authors response to my comment: We appreciate the reviewers comment. We honestly dont see how FTIR can be used in vivo. However, we believe that it can be a good tool to aid diagnosing NAFLD in liver biopsies. Currently definitive NAFLD diagnosis is based on histology performed by pathologists. FTIR could be a quick technique to assist identify structural changes that reflect NAFLD stages. It can also be used to monitor response to treatments. We have changed the conclusion to include this information.  I suggest that the authors remove the word noninvasive from the manuscript because noninvasive method is more meaningful to in vivo diagnosis.

4. In the authors response to my comment: Thank you for the comment. We agree that the effect of EX-1 treatment on the structural changes of lipids and proteins is limited. However, FTIR is still able to detect these changes, which reflects the high sensitivity of the technique. It also indicates that small changes in the structures of proteins or lipids could have a huge impact on the cellular function. I agree that FTIR is still able to detect these changes, which reflects the high sensitivity of the technique, however, my comment means that compared with the changes between control group and OA treatment, the changes between OA treatment and OA+EX-1 treatment were quite small, which indicate that the effect of EX-1 on NAFLD is not significant, especially in terms of structural changes of lipids and proteins. I only see a big reduction of OA-induced lipid accumulation by EX-1 treatment. I suggest that the authors be cautious to make claims on the effect of EX-1 treatment. Please carefully check  and revise the manuscript when they overstate the effect of EX-1. 

Author Response

Dear Reviewer,

We really appreciate your effort and time to help us improve the quality of our manuscript. To the best of our ability, we have addressed the remaining concerns that you have raised. Please see the attachment.

Best regards
